# Improving Hierarchical Adversarial Robustness of Deep Neural Networks

## Abstract

Do all adversarial examples have the same consequences? An autonomous driving system misclassifying a pedestrian as a car may induce a far more dangerous —and even potentially lethal—behavior than, for instance, a car as a bus. In order to better tackle this important problematic, we introduce the concept of *hierarchical* adversarial robustness. Given a dataset whose classes can be grouped into *coarse-level* labels, we define hierarchical adversarial examples as the ones leading to a misclassification at the coarse level. To improve the resistance of neural networks to hierarchical attacks, we introduce a *hierarchical adversarially robust* (HAR) network design that decomposes a single classification task into one coarse and multiple fine classification tasks, before being specifically trained by adversarial defense techniques. As an alternative to an end-to-end learning approach, we show that HAR significantly improves the robustness of the network against $\ell_2$ and $\ell_\infty$ bounded hierarchical attacks on the CIFAR-100 dataset.

## 1 Introduction

Deep neural networks (DNNs) are highly vulnerable to attacks based on small modification of the input to the network at test time (Szegedy et al., 2013). Those adversarial perturbations are carefully crafted in a way that they are imperceptible to human observers, but when added to clean images, can severely degrade the accuracy of the neural network classifier. Since their discovery, there has been a vast literature proposing various attack and defence techniques for the adversarial settings (Szegedy et al., 2013; Goodfellow et al., 2014; Kurakin et al., 2016; Madry et al., 2017; Wong et al., 2020). These methods constitute important first steps in studying adversarial robustness of neural networks. However, there exists a fundamental flaw in the way we assess a defence or an attack mechanism. That is, we overly generalize the mistakes caused by attacks. Particularly, the current approaches focuses on the scenario where different mistakes caused by the attacks are treated equally. We argue that some context do not allow mistakes to be considered equal. In CIFAR-100 (Krizhevsky et al., 2009), it is less problematic to misclassify a pine tree as a oak tree than a fish as a truck.

As such, we are motivated to propose the concept of hierarchical adversarial robustness to capture this notion. Given a dataset whose classes can be grouped into coarse labels, we define hierarchical adversarial examples as the ones leading to a misclassification at the coarse level; and we present a variant of the projected gradient descent (PGD) adversaries (Madry et al., 2017) to find hierarchical adversarial examples. Finally, we introduce a simple and principled hierarchical adversarially robust (HAR) network which decompose the end-to-end robust learning task into a single classification task into one coarse and multiple fine classification tasks, before being trained by adversarial defence techniques. Our contribution are

- We introduce the concept of hierarchical adversarial examples: a special case of the standard adversarial examples which causes mistakes at the coarse level (Section 2).
- We present a *worst-case* targeted PGD attack to find hierarchical adversarial examples. The attack iterates through all candidate fine labels until a successful misclassification into the desired target (Section 2.1).
- We propose a novel architectural approach, HAR network, for improving the hierarchical adversarial robustness of deep neural networks (Section 3). We empirically show that HAR networks significantly improve the hierarchical adversarial robustness against $\ell_\infty$ attacks ($\epsilon = \frac{8}{255}$) (Section 4) and $\ell_2$ attacks ($\epsilon = 0.5$) (Appendix A.4) on CIFAR-100.

- We benchmark using untargeted PGD20 attacks as well as the proposed iterative targeted PGD attack. In particular, we include an extensive empirical study on the improved hierarchical robustness of HAR by evaluating against attacks with varying PGD iterations and $\epsilon$. We find that a vast majority of the misclassifications from the untargeted attack are within the same coarse label, resulting a failed hierarchical attack. The proposed iterative targeted attacks provides a better empirical representation of the hierarchical adversarial robustness of the model (Section 4.2).

- We show that the iterative targeted attack formulated based on the coarse network are weaker hierarchical adversarial examples compared to the ones generated using the entire HAR network (Section 4.3).

## 2 HIERARCHICAL ADVERSARIAL EXAMPLES

The advancement in DNN image classifiers is accompanied by the increasing complexity of the network design (Szegedy et al., 2016; He et al., 2016), and those intricate networks has provided state-of-the-art results on many benchmark tasks (Deng et al., 2009; Geiger et al., 2013; Cordts et al., 2016; Everingham et al., 2015). Unfortunately, the discovery of adversarial examples has revealed that neural networks are extremely vulnerable to maliciously perturbed inputs at test time (Szegedy et al., 2013). This makes it difficult to apply DNN-based techniques in mission-critical and safety-critical areas.

Another important develeopment along with the advancement of DNN is the growing complexity of the dataset, both in size and in number of classes: i.e. from the 10-class MNIST dataset to the 1000-class ImageNet dataset. As the complexity of the dataset increases exponentially, dataset can often be divided into several coarse classes where each coarse class consists of multiple fine classes. In this paper, we use the term label and class interchangeably.

The concept of which an input image is first classified into coarse labels and then into fine labels are referred to as *hierarchical classification* (Tousch et al., 2012). Intuitively, the visual separability between groups of fine labels can be highly uneven within a given dataset, and thus some coarse labels are more difficult to distinguish than others. This motivates the use of more dedicated classifiers for specific groups of classes, allowing the coarse labels to provide information on similarities between the fine labels at an intermediate stage. The class hierarchy can be formed in different ways, and it can be learned strategically for optimal performance of the downstream task (Deng et al., 2011). Note that it is also a valid strategy to create a customized class hierarchy and thus be able to deal with sensitive missclassification. To illustrate our work, we use the predefined class hierarchy of the CIFAR-10 and the CIFAR-100 dataset (Krizhevsky et al., 2009): fine labels are grouped into coarse labels by semantic similarities.

All prior work on adversarial examples for neural networks, regardless of defences or attacks, focuses on the scenario where all misclassifications are considered equally (Szegedy et al., 2013; Goodfellow et al., 2014; Kurakin et al., 2016; Madry et al., 2017; Wong et al., 2020). However, in practice, this notion overly generalizes the damage caused by different types of attacks. For example, in an autonomous driving system, confusing a perturbed image of a traffic sign as a pedestrian should not be treated the same way as confusing a bus as a pickup truck. The former raises a major security threat for practical machine learning applications, whereas the latter has very little impact to the underlying task. Moreover, misclassification across different coarse labels poses potential ethical concerns when the dataset involves sensitive features such as different ethnicities, genders, people with disabilities and age groups.

> *Mistakes across coarse classes leads to much more severe consequences compared to mistakes within coarse classes.*

As such, to capture this different notion of attacks, we propose the term *hierarchical* adversarial examples. They are a specific case of adversarial examples where the resulting misclassification occurs between fine labels that come from different coarse labels.

Here, we provide a clear definition of the hierarchical adversarial examples to differentiate it from the standard adversarial examples. We begin with the notation for the classifier. Consider a neural network $F(x) : \mathbb{R}^d \to \mathbb{R}^c$ with a softmax as its last layer (Hastie et al., 2009), where $d$ and $c$

denote the input dimension and the number of classes, respectively. The prediction is given by $\arg\max_i F(x)_i$.

In the hierarchical classification framework, the classes are categorized (e.g. by the user) into fine classes and coarse classes [1]. The dataset consists of image and fine label pairs: $\{x, y\}_n$. In the later, we use the set theory symbol $\in$ to characterize the relationship between a fine and a coarse label: $y \in z$ if the fine label $y$ is part of the coarse class $z$. Note that this relation holds for both disjoint and overlapping coarse classes. Given an input data $x$, suppose its true coarse and fine labels are $z^*$ and $y^*$ respectively. Under the setting defined above, a hierarchical adversarial example must satisfy all the following properties:

- the unperturbed input data $x$ is correctly classified by the classifier: $\arg\max_i F(x)_i = y^*$;
- the perturbed data $x' = x + \delta$ is perceptually indistinguishable from the original input $x$;
- the perturbed data $x'$ is classified incorrectly: $\arg\max_i F(x')_i = y'$ where $y' \neq y^*$;
- the misclassified label belongs to a different coarse class: $y' \notin z^*$.

Notice that satisfying the first three properties is sufficient to define a standard adversarial examples, and that hierarchical adversarial examples are special cases of adversarial examples. It is worth mentioning that measuring perceptual distance can be difficult (Li et al., 2003), thus the second property is often replaced by limiting that the adversary can only modify any input $x$ to $x + \delta$ with $\delta \in \Delta$. Commonly used constraints are $\epsilon$-balls w.r.t. $\ell_p$-norms, though other constraint sets have been used too (Wong et al., 2019). In this work, we focus on $\ell_\infty$- and $\ell_2$-norm attacks.

### 2.1 GENERATING HIERARCHICAL ADVERSARIAL PERTURBATIONS

A common class of attack techniques are gradient-based attacks, such as FGSM (Goodfellow et al., 2014), BIM (Kurakin et al., 2016) and PGD (Madry et al., 2017), that utilize gradient (first-order) information of the network to compute perturbations. Such methods are motivated by linearizing the loss function and solving for the perturbation that optimizes the loss subject to the $\ell_p$-norm constraint. Their popularity is largely due to its simplicity, because the optimization objective can be accomplished in closed form at the cost of one back-propagation.

The main idea of gradient-based attacks can be summarized as follows. Given the prediction of $F(x)$ and a target label $y$, the loss function of the model is denoted by $\ell(x, y) \triangleq \ell(F(x), y)$, e.g., a cross-entropy loss. Here, we omit the network parameter $w$ in the loss because it is assumed to be fixed while generating adversarial perturbations. Note that the choice of $y$ and whether to maximize or minimize the loss depend on if the attack is targeted or untargeted. For a targeted $\ell_\infty$ attack, gradient-based methods rely on the opposite direction of the loss gradient, $-\operatorname{sign}\nabla_x \ell(x, y)$, to solve for the perturbation that **minimizes** the loss with respect to a non-true target label ($y \neq y^*$). Despite its simplicity, gradient-based attacks are highly effective at finding $\ell_p$-bounded perturbations that lead to misclassifications.

In our work, we introduce a simple variant of the projected gradient descent (PGD) adversary to find hierarchical adversarial examples. Given an input image with true coarse and fine labels $z^*$ and $y^*$ respectively. Let $x_j$ denote the perturbed input at iteration $j$, we define:

$$x_{j+1} = \Pi_{B_\infty(x, \epsilon)} \left\{ x_j - \alpha \operatorname{sign}\left(\nabla_x \ell(x_j, \hat{y})\right) \right\} \tag{1}$$

where the target label $\hat{y}$ comes from a different coarse class: $\hat{y} \notin z^*$. Algorithm 1 summarize the procedures for generating an $\ell_\infty$-constrained hierarchical adversarial examples. The projection operator $\Pi$ after each iteration ensures that the perturbation is in an $\epsilon$-neighbourhood of the original image. We also adopt the random initialization in PGD attacks (Madry et al., 2017): $x_0 = x + \eta$, where $\eta = (\eta_1, \eta_2, \ldots, \eta_d)^\top$ and $\eta_i \sim \mathcal{U}(-\epsilon, \epsilon)$.

There are several approaches to choose the target class (Carlini & Wagner, 2017). The target class can be chosen in an *average-case* approach where the class is selected uniformly at random among all eligible labels. Alternatively, they can be chosen in a strategic way, a *best-case* attack, to find the target class which requires the least number of PGD iterations for misclassifications. In our work, we consider a *worst-case* attack by iterating through all candidate target labels, i.e., fine labels that

---

[1]We could go beyond this 2-level hierarchy. Here we keep the presentation simple for didactic purposes.

---

**Algorithm 1:** A *worst-case* approach for generating $\ell_\infty$-bounded hierarchical adversarial example based on a targeted PGD attack.

---

**Input** : A pair of input data $(x, y^*)$, where fine label $y^*$ belongs to the coarse label $z^*$; a neural network $F(\cdot)$; loss function $\ell(\cdot)$; $\ell_\infty$ constraint of $\epsilon$; number of PGD iterations $k$; PGD step-size $\alpha$.

1 Define $S = \{y \mid y \notin z^*\}$, a collection of all fine labels that do not belong in the coarse label $z^*$;
2 **for** $\hat{y} \in S$ **do**
3     $x_0 \leftarrow x + \eta$, where $\eta \leftarrow (\eta_1, \eta_2, \ldots, \eta_d)^\top$ and $\eta_i \sim \mathcal{U}(-\epsilon, \epsilon)$.
4     **for** $j = 0, \ldots, k-1$ **do**
5        $x_{j+1} = \Pi_{B_\infty(x,\epsilon)} \{x_j - \alpha \operatorname{sign}(\nabla_x \ell(x_j, \hat{y}))\}$ where $\Pi$ is the projection operator.
6     **end**
7     **if** $\arg\max_i F(x_k)_i = \hat{y}$ **then**
8        Terminate (successful attack);
9     **else**
10        $S \setminus \hat{y}$;
11        **if** $S$ *is empty* **then**
12           Terminate (failed attack);
13        **end**
14     **end**
15 **end**

---

do not belong in the same coarse class. This iterative targeted attack process terminates under two conditions: 1. perturbation results in a successful targeted misclassification; 2. all candidate fine labels have been used as targets.

## 2.2 RELATED WORK ON HIERARCHICAL CLASSIFICATION

In image classification domain, there is a sizable body of work exploiting class hierarchy of the dataset (Tousch et al., 2012). For classification with a large number of classes, it is a common technique to divide the end-to-end learning task into multiple classifications based on the semantic hierarchy of the labels (Marszałek & Schmid, 2008; Liu et al., 2013; Deng et al., 2012). They are motivated by the intuition that some coarse labels are more difficult to distinguish than others, and specific category of classes requires more dedicated classifiers. A popular hierarchy structure is to divide the fine labels into a label tree with root nodes and leaf nodes. Deng et al. (2011) propose an efficient technique to simultaneously determine the structure of the tree as well as learning the classifier for each node in the tree. Instead of learning the optimal tree structure, it is also common to use the predefined hierarchy of the dataset Deng et al. (2012).

## 3 HIERARCHICAL ADVERSARIALLY ROBUST (HAR) NETWORK

To improve the hierarchical adversarial robustness of neural networks, we propose a simple and principled hierarchical adversarially robust (HAR) network which decompose the end-to-end robust learning task into two parts. First, we initialize a neural network for the coarse classification task, along with multiple networks for the fine classification tasks. Next, all the networks are trained using adversarial defence techniques to improve the robustness of their task at hand. The final probability distribution of all the fine classes are computed based on Bayes Theorem. For brevity, we use coarse neural network (CNN) and fine neural network (FNN) to denote the two different types of networks. Intuitively, the HAR network design benefits from a single robustified CNN with improved robustness between coarse classes, and multiple robustified FNN with improved the robustness between visually similar fine classes.

### 3.1 ARCHITECTURE DESIGN OF HAR

Instead of the traditional flat design of the neural network, HAR consists of one CNN for the coarse labels and several FNNs for the fine labels. Note that there is a one-to-one correspondence between

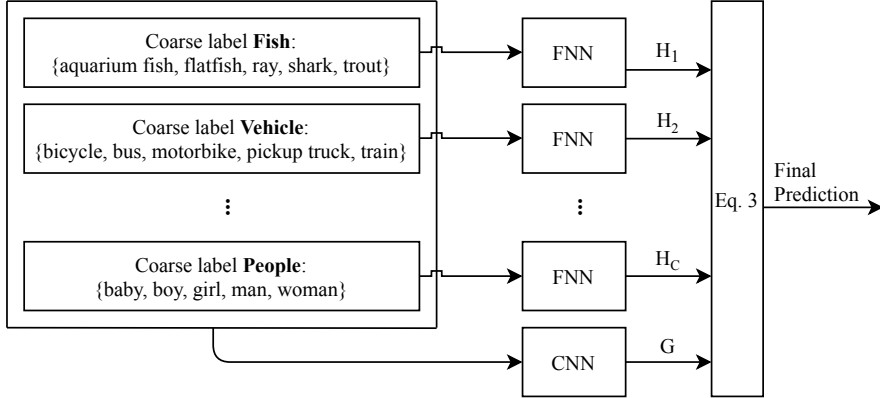

Figure 1: Pipeline of the proposed HAR network design to improve hierarchical adversarial robustness of the neural network.

a particular FNN and a specific group of fine labels. Such a module design mimics the hierarchical structure of the dataset where the fine classes are grouped into coarse classes. Recall that our definition of the neural network $F(x)$ includes the softmax function as its last layer, so the output of the network can be interpreted as the probability distribution of the classes: $P(y \mid x)$. Conditioned on the probability of the coarse class, we can define the fine class probability as

$$P(y \mid x) = P(y \mid x, z)P(z \mid x). \tag{2}$$

Here, the probability distribution of the fine classes are computed as the product of two terms. Given an input $x$, the first term $P(y \mid x, z)$ represents the probability of $x$ being a fine label $y$ where $y$ belongs to a coarse class $z$. This is essentially FNNs' prediction of the fine classes within a coarse category. The second term $P(z \mid x)$ represents the probability of $x$ being a coarse label $z$, and it can be understood as the prediction from the CNN. With this decomposition of the original learning task, we can reconstruct the fine label distribution by probabilistically combining the predictions from the different networks.

An important advantage of this flexible modular network design is that it allows us to train the component networks using adversarial defence techniques to improve the robustness of their associated task. Especially, a robustified coarse neural network leads to improved hierarchical adversarial robustness between coarse labels. During training, each component of the HAR network is trained independently, allowing them to be trained in parallel. We use the entire dataset with the coarse labels, $\{x, z\}$, to train the coarse class network $G(x)$, followed by training multiple fine class network $H(x)$ using only a portion of the dataset.

The inference procedure can be described as follows. Suppose the number of coarse classes in a dataset is $C$, and each coarse class contains $j$ number of fine classes. Similar to the definition of $F(x)$, we use $G(x)$ to denote the output of the CNN: $G(x) = [g_1, \ldots, g_c]$. We use $H_i(x)$ to denote the output of the FNN: $H_i(x) = \left[h_1^i, \ldots, h_j^i\right]$, where $j$ is an positive integer indicating the number of fine classes in the coarse class $i$. In this setting, the output of the combined neural network is:

$$F(x) = [g_1 H_1(x), \ldots, g_C H_C(x)]. \tag{3}$$

### 3.2 Related Work on Adversarial Defence methods

A plethora of defence mechanisms have been proposed for the adversarial setting. Adversarial training (Szegedy et al., 2013) is one of the standard approaches for improving the robustness of deep neural networks against adversarial examples. It is a data augmentation method that replaces unperturbed training data with adversarial examples and updates the network with the replaced data points. Intuitively, this procedure encourages the DNN not to make the same mistakes against an adversary. By adding sufficiently enough adversarial examples, the network gradually becomes robust to the attack it was trained on. Existing adversarial training methods (Szegedy et al., 2013; Goodfellow et al., 2014; Kurakin et al., 2016; Madry et al., 2017; Wong et al., 2020) differ in

the adversaries used in the training. Another related line of adversarial defence methods focuses on regularizing the loss function instead of data augmentation. TRADES (Zhang et al., 2019) introduces a regularization term that penalizes the difference between the output of the model on a training data and its corresponding adversarial example. The regularized loss consists of a standard cross-entropy loss on the unperturbed data and a KL-divergence term measuring the difference between the distribution of clean training data and adversarially perturbed training data.

## 4 EXPERIMENTS

In this section, we evaluate the hierarchical adversarial robustness of the HAR network design, incorporating two popular adversarial defence methods: adversarial training with PGD10 adversaries (Madry et al., 2017) and TRADES (Zhang et al., 2019). In this section, we focus on evaluations based on $\ell_\infty$ norm attacks, and defer evaluations on $\ell_2$ in Appendix A.4. Compared to the traditional flat design of neural network, our experiments show that HAR leads to a significant improvement in hierarchical adversarial robustness under various targeted and untargeted $\ell_\infty$ attacks.

### 4.1 EVALUATION SETUP

We use network architectures from the ResNet family (He et al., 2016) on the CIFAR-100 dataset. The hierarchical structure of classes within the two dataset is illustrated in Table 6. To establish a baseline, we train ResNet50 networks using the following methods: (1) Standard: training with unperturbed data; (2) ADV: training with 10-step untargeted PGD examples (2) ADV-T: training with 10-step randomly targeted PGD examples and (4) TRADES. ADV-T is a targeted-version of the PGD adversarial training. Specifically, given an input pair from the training set $(x, y)$ and $y \in z^*$, the perturbation is computed based on a targeted 10-step PGD attack where the target label is uniformly random sampled from $\{y \mid y \notin z^*\}$. We refer the flat models as *vanilla* models. For training HAR networks, we use ResNet10 for both the coarse network and the fine network. We use models with a lower capacity to reduce the difference in the order of magnitude of parameters between a single ResNet50 model and multiple ResNet10 models. This is to eliminate the concern of which the improved hierarchical adversarial robustness is obtained due to the increasing network complexity. A comparison between the number of trainable parameters is included in Appendix A.1. Note that in the HAR network, all component networks (CNN and FNNs) are trained using the same adversarial defence approach. As a concrete example, a HAR network trained with TRADES on CIFAR100 consists of one coarse classifier and twenty fine classifiers where they are all trained using TRADES.

For all four methods (Standard, ADV, ADV-T and TRADES), networks are trained for a total of 200 epochs, with an initial learning rate of 0.1. The learning rate decays by an order of magnitude at epoch 100 and 150. We used a minibatch size of 128 for testing and training. We used SGD optimizer with momentum of 0.9 and a weight decay of 2e-4. For TRADES, we performed a hyperparameter sweep on the strength of the regularization term $\beta$ and selected one that resulted in the highest accuracy against untargeted $\ell_\infty$ bounded PGD20 attacks. The optimization procedure is used for both the vanilla models and all component models in the HAR network.

### 4.2 HIERARCHICAL ROBUSTNESS UNDER UNTARGETED AND TARGETED ATTACKS

There are several threat models to consider while evaluating adversarial robustness, regardless of standard or hierarchical robustness. The white-box threat model specifies that the model architecture and network parameters are fully transparent to the attacker (Goodfellow et al., 2014). Despite many white-box attack methods exist, perturbations generated using iterations of PGD remain as one of the most common benchmarks for evaluating adversarial robustness under the white-box setting. As such, we use PGD as the main method to generate both untargeted and targeted attacks. Specifically, we perform 20, 50, 100 and 200 iterations of PGD for the untargeted attacks in Table 1. Due to the large number of fine labels in CIFAR-100, we randomly selected 1000 test set input and perform the iterative, worst-case hierarchical adversarial perturbations introduced in Section 2.1. Note that the evaluation on untargeted attacks uses the entire test set. The results on the worst-case targeted attack is included in Table 2. Note that, besides $\epsilon = 8/255$, we also evaluated HAR against attacks with

Table 1: Accuracy of different models on CIFAR100 against $\ell_\infty$ bounded white-box untargeted PGD attacks. (a higher score indicates better performance)

| Method | | Clean | | PGD20 ($\epsilon = \frac{4}{255}$) | | PGD20 ($\epsilon = \frac{8}{255}$) | |
|---|---|---|---|---|---|---|---|
| | | Fine | Coarse | Fine | Coarse | Fine | Coarse |
| Vanilla | Standard | 73.21% | 82.57% | 0.01% | 24.89% | 0.00% | 20.13% |
| | ADV | 58.62% | 69.81% | 21.36% | 37.80% | 11.56% | 29.06% |
| | ADV-T | 64.74% | 75.02% | 17.19% | 41.30% | 7.60% | 34.74% |
| | TRADES | 57.12% | 67.67% | 26.69% | 41.47% | 17.32% | 32.36% |
| HAR | Standard | 63.49% | 81.24% | 0.12% | 29.25% | 0.00% | 22.63% |
| | ADV | 48.53% | 66.23% | 20.28% | 30.53% | 11.64% | 20.71% |
| | TRADES | 46.62% | 62.49% | 22.00% | 32.86% | 14.29% | 22.59% |
| Method | | PGD50 ($\epsilon = \frac{8}{255}$) | | PGD100 ($\epsilon = \frac{8}{255}$) | | PGD200 ($\epsilon = \frac{8}{255}$) | |
| | | Fine | Coarse | Fine | Coarse | Fine | Coarse |
| Vanilla | Standard | 0.01% | 24.84% | 0.00% | 25.12% | 0.01% | 24.94% |
| | ADV | 21.05% | 37.16% | 20.87% | 36.96% | 20.94% | 37.02% |
| | ADV-T | 16.94% | 41.07% | 16.88% | 40.99% | 16.79% | 40.84% |
| | TRADES | 26.48% | 41.22% | 26.58% | 41.40% | 26.52% | 41.27% |
| HAR | Standard | 0.15% | 28.91% | 0.14% | 29.46% | 0.15% | 29.27% |
| | ADV | 19.91% | 29.94% | 20.03% | 29.99% | 19.89% | 29.93% |
| | TRADES | 21.99% | 32.66% | 21.90% | 32.60% | 21.87% | 32.36% |

Table 2: Accuracy of different models on CIFAR100 against $\ell_\infty$ bounded worst-case targeted PGD attacks generated based on Algorithm 1. (a higher score indicates better performance)

| Method | | PGD20 ($\epsilon = \frac{4}{255}$) | PGD20 ($\epsilon = \frac{8}{255}$) | PGD50 ($\epsilon = \frac{8}{255}$) | PGD100 ($\epsilon = \frac{8}{255}$) | PGD200 ($\epsilon = \frac{8}{255}$) |
|---|---|---|---|---|---|---|
| Vanilla | Standard | 0.00% | 0.00% | 0.00% | 0.00% | 0.00% |
| | ADV | 43.30% | 24.60% | 24.60% | 24.50% | 24.00% |
| | ADV-T | 43.50% | 22.10% | 21.70% | 20.70% | 21.00% |
| | TRADES | 47.20% | 30.00% | 29.80% | 29.70% | 28.80% |
| HAR | Standard | 8.60% | 4.00% | 3.50% | 3.40% | 3.30% |
| | ADV | **43.70%** | **25.80%** | **25.50%** | **25.30%** | **24.00%** |
| | TRADES | 45.80% | 29.20% | 28.90% | 29.30% | 28.90% |

$\epsilon = 4/255$. All $\ell_\infty$-PGD adversarial examples used for all evaluations are generated a step size of $\epsilon/4$ (pixel values are normalized to $[0, 1]$).

Along with the two attacks, we also include results on unperturbed testset data (Clean). For clean and untargeted attacks, we report the percentage of correct fine class prediction as fine accuracy, and the percentage of fine class prediction belonging to the correct coarse class as coarse accuracy. For targeted, its accuracy refers to the percentage of the testset data where the targeted attack fails to alter the final prediction to the desired target, even after iterating through all eligible target labels. It is important to realize that a successful targeted attack implies misclassification for both coarse and fine classes. Table 1 summarize the accuracy of the HAR model and vanilla models on standard unperturbed data, and against untargeted and targeted attacks.

Table 3: Accuracy of the hierarchical classifier on CIFAR-100 against $\ell_\infty$ bounded targeted attacks ($\epsilon = 8/255$) generated using the coarse network (Coarse). As a comparison, the attack counterpart generated using the entire HAR network is also included (HAR). (a higher score indicates better performance)

| Method | Coarse | HAR |
|---|---|---|
| Standard | 0.00% | 3.30% |
| ADV | 29.96% | 24.00% |
| TRADES | 29.38% | 28.00% |

### 4.2.1 DISCUSSIONS

Before making the comparison between the HAR model and the vanilla model, we make an interesting observation: untargeted attacks often results in misclassification within the same coarse class, shown by the high coarse accuracy under Untargeted. In particular, vanilla networks trained with unperturbed training data have $0\%$ fine accuracy under Untargeted, while a vast majority of the misclassified classes still belong to the correct coarse class. This shows that the untargeted attacks do not provide a good representation of the hierarchical adversarial robustness as an empirical evaluation. On the other hand, the iterative targeted attack leads to a severe damage in hierarchical adversarial robustness for vanilla models trained with all three methods. On Standard trained models, despite the high hierarchical robustness under untargeted attacks, nearly all of the CIFAR10 and CIFAR100 testset data can be perturbed into a desired target class from another coarse class. As such, we emphasize the use of the iterative worst-case targeted attack for a more accurate evaluations for the hierarchical adversarial robustness of the model. For vanilla models trained with ADV and TRADES, we notice that the improved adversarial robustness on fine classes also translates to an improvement on hierarchical adversarial robustness. We observed that vanilla models trained using ADV-T shows improved robustness against untargeted PGD attacks compared to the original adversarial training method. However, we noticed that there is a significant decrease in the hierarchical robustness of ADV-T models against the hierarchical adversarial examples, leading to a worse hierarchical robust accuracy compared to ADV. Finally, we observe that HAR network trained with ADV and TRADES significantly improves the robustness against iterative targeted attacks compared to the vanilla counterparts. On CIFAR-100, HAR network achieves a $1.2\%$ improvement on PGD20 adversaries ($\epsilon = 8/255$) when trained with ADV.

### 4.3 HIERARCHICAL ROBUSTNESS UNDER TARGETED ATTACKS BASED ON THE COARSE NETWORK

Under the white-box threat model, attackers with a complete knowledge of the internal structure of HAR can also generate perturbations based on the coarse network. During evaluations, we investigate whether the targeted PGD adversaries based on the coarse network are stronger hierarchical adversarial examples compared to the ones generated using the entire network. Such attacks can be understood as finding a more general perturbation which alters the probability distribution of the coarse class: $P(z \mid x)$. Similar to the attack proposed in Section 2.1, we perform an iterative, worst-case targeted PGD20 attack based on the coarse neural network. Specifically, we replace $\ell(F(x), y)$ with $\ell(G(x), z)$ in Eq. 1, and iterate through all eligible coarse classes as target labels. For example, to generate such attacks for HAR with ADV-trained component networks, the iterative targeted attack is performed based on the ADV-trained coarse network in the original HAR network. Note that there is a distinction between the above attack procedure and a transfer-based attack where the perturbation is transferred from an independently trained source model (Papernot et al., 2017). Since the perturbation is generated using part of the HAR network, such attacks still belongs in the white-box setting. Our results in Table 3 show that the perturbations generated using the coarse network are weaker attacks compared to the ones generated using the entire network.

## 5 CONCLUSION

In this work, we introduced a novel concept called hierarchical adversarial examples. For dataset which classes can be further categorized into fine and coarse classes, we defined hierarchical adversarial examples as the ones leading to a misclassication at the coarse level. To improve the hierarchical adversarial robustness of the neural network, we proposed the HAR network design, a composite of a coarse network and fine networks where each component network is trained independently by adversarial defence techniques. We empirically showed that HAR leads to a significant increase in hierarchical adversarial robustness under white-box untargeted/targeted attacks on CIFAR-10 and CIFAR-100-5x5.

The rapid adoption of machine learning applications has also led to an increasing importance in improving the robustness and reliability of such techniques. Mission-critical and safety-critical systems which rely on DNN in their decision-making process shall incorporate robustness, along with accuracy, in their development process. The introduction of the hierarchical adversarial examples and ways to defend against them is an important step towards a more safe and trustworthy AI system.

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

# A APPENDIX

## A.1 COMPARISON OF TRAINABLE MODEL PARAMETERS

In our evaluations, we use ResNet50 for the vanilla models, and use multiple ResNet10 for the HAR network. We use models with a lower capacity to reduce the difference in the order of magnitude of parameters between a single ResNet50 model and multiple ResNet10 models. This helps to address the concern of which the improved hierarchical adversarial robustness is obtained due to the increasing network complexity.

Table 4: Number of trainable parameters in ResNet10 and ResNet34

| Model | Number of Parameters |
|-------|---------------------|
| ResNet10 | 4.9 M |
| ResNet50 | 23.3 M |

## A.2 HYPERPARAMETER SWEEP FOR TRADES ON RESNET10

The following results show the hyperparameter sweep on TRADES. We include the one with the highest PGD20 accuracy in Section 4.

Table 5: Hyperparameter sweep of TRADES on ResNet10: evaluation based performance on CIFAR-10 against $\ell_\infty$ bounded adversarial perturbations ($\epsilon = 8/255$).

| $\beta$ | Standard | FGSM | PGD20 |
|---|---|---|---|
| 15 | 72.80% | 48.85% | 44.80% |
| 13 | 73.46% | 49.18% | 45.09% |
| 11 | 74.46% | 49.39% | 44.86% |
| 9 | 75.61% | **50.06**% | **45.38**% |
| 7 | 76.74% | 50.05% | 45.04% |
| 5 | 78.39% | 50.22% | 44.40% |
| 3 | 80.20% | 49.84% | 42.53% |
| 1 | 83.69% | 45.61% | 35.27% |
| 0.5 | 84.00% | 46.16% | 34.40% |
| 0.1 | 84.64% | 41.12% | 16.52% |

## A.3    HIERARCHICAL STRUCTURE OF CLASSES WITHIN THE CIFAR10 DATASET

Table 6: The hierarchical structure of classes within the CIFAR-10 and CIFAR-100 dataset

|            | Coarse labels                   | Fine labels                                            |
|------------|---------------------------------|--------------------------------------------------------|
| CIFAR-10   | Animals                         | bird, cat, deer, dog, frog, horse                      |
|            | Vehicles                        | airplane, automobile, ship, truck                      |
| CIFAR-100  | Aquatic mammals                 | beaver, dolphin, otter, seal, whale                    |
|            | Fish                            | aquarium fish, flatfish, ray, shark, trout             |
|            | Flowers                         | orchids, poppies, roses, sunflowers, tulips            |
|            | Food containers                 | bottles, bowls, cans, cups, plates                     |
|            | Fruit and vegetables            | apples, mushrooms, oranges, pears, sweet peppers       |
|            | Household electrical devices    | clock, computer keyboard, lamp, telephone, television  |
|            | Household furniture             | bed, chair, couch, table, wardrobe                     |
|            | Insects                         | bee, beetle, butterfly, caterpillar, cockroach         |
|            | Large carnivores                | bear, leopard, lion, tiger, wolf                       |
|            | Large man-made outdoor things   | bridge, castle, house, road, skyscraper                |
|            | Large natural outdoor scenes    | cloud, forest, mountain, plain, sea                    |
|            | Large omnivores and herbivores  | camel, cattle, chimpanzee, elephant, kangaroo          |
|            | Medium-sized mammals            | fox, porcupine, possum, raccoon, skunk                 |
|            | Non-insect invertebrates        | crab, lobster, snail, spider, worm                     |
|            | People                          | baby, boy, girl, man, woman                            |
|            | Reptiles                        | crocodile, dinosaur, lizard, snake, turtle             |
|            | Small mammals                   | hamster, mouse, rabbit, shrew, squirrel                |
|            | Trees                           | maple, oak, palm, pine, willow                         |
|            | Vehicles 1                      | bicycle, bus, motorcycle, pickup truck, train          |
|            | Vehicles 2                      | lawn-mower, rocket, streetcar, tank, tractor           |

## A.4 RESULTS ON CIFAR100 WITH $\ell_2$ ATTACKS

Overall, we observe a similar robustness improvement with the HAR network. Note that we were not able to achieve reasonable robustness results with TRADES against $\ell_2$ attacks. One possible reason is that the hyper-parameter sweet was based on the $\ell_\infty$ results, as such not suitable for $\ell_2$ attacks. For this reason, we omit TRADES in this section.

Table 7: Accuracy of different models on CIFAR100 against $\ell_2$ bounded white-box untargeted PGD attacks. (a higher score indicates better performance)

| Method | | Clean | | PGD20 ($\epsilon = 0.5$) | | PGD20 ($\epsilon = 0.25$) | |
|---|---|---|---|---|---|---|---|
| | | Fine | Coarse | Fine | Coarse | Fine | Coarse |
| Vanilla | Standard | 73.21% | 82.57% | 0.29% | 31.86% | 3.05% | 35.95% |
| | ADV | 64.38% | 74.85% | 36.35% | 53.77% | 50.04% | 64.05% |
| | ADV-T | 68.73% | 78.58% | 31.41% | 53.10% | 48.78% | 64.62% |
| HAR | ADV | 56.96% | 73.31% | 31.91% | 48.94% | 43.87% | 60.68% |

| Method | | PGD50 ($\epsilon = 0.5$) | | PGD100 ($\epsilon = 0.5$) | | PGD200 ($\epsilon = 0.5$) | |
|---|---|---|---|---|---|---|---|
| | | Fine | Coarse | Fine | Coarse | Fine | Coarse |
| | Standard | 0.17% | 32.17% | 0.13% | 32.55% | 0.12% | 32.07% |
| Vanilla | ADV | 36.16% | 53.64% | 36.11% | 53.61% | 36.07% | 53.59% |
| | ADV-T | 31.21% | 52.97% | 31.15% | 52.82% | 31.09% | 52.78% |
| HAR | ADV | 31.90% | 48.78% | 31.76% | 48.70% | 31.70% | 48.91% |

Table 8: Accuracy of different models on CIFAR100 against $\ell_2$ bounded worst-case targeted PGD attacks generated based on Algorithm 1. (a higher score indicates better performance)

| Method | | PGD20 ($\epsilon = 0.5$) | PGD50 ($\epsilon = 0.5$) |
|---|---|---|---|
| | Standard | 0.10% | 0.10% |
| Vanilla | ADV | 43.60% | 42.90% |
| | ADV-T | 39.20% | 38.50% |
| HAR | ADV | **44.30%** | **44.00%** |

