# OpenReview forum: "Improving Hierarchical Adversarial Robustness of Deep Neural Networks"
_ICLR.cc/2021/Conference — Reject_

### Official Review · AnonReviewer3 · 2020-10-23
**more experiments necessary: no comparisons to sota, no ablations**

**Rating:** 5
**Confidence:** 4

**Review:**

The authors present an interesting hierarchical decomposition of cifar classification and show that it helps the model perform better against adversarial robustness.

There are some general issues with the paper. For starters, it is usual in benchmarking to show a plot of accuracy against PGD steps. This is done because eventually, accuracy should stabilize somewhere. It is not clear why the authors settle on 20 steps. While the authors show HAR gives an improvement against the vanilla model at 20 steps, this might not hold if the number of attack steps were increased.

I am not sure how this method's cifar accuracies compare to sota. The authors should show an improvement against the state of the art on cifar-10. Otherwise, it is difficult to understand what is the value of including this two-step training procedure.

I would also like to see how sensitive the coarse model is to the choice of z input; an ablation study should be performed. One possible suggestion for this would be to train two models, p(z|x) and p(y|x,z) but where z is just the repeated, fine-grained y label. This would tell us what is the amount of improvement coming simply from having a two-step classification process involving different neural net architectures.

---

> ### Author Response · Authors · 2020-11-25
> **Response to the Reviewer**
>
> Thank you for the careful reading of the paper and your helpful comments. We revised the paper with an extensive empirical study on CIFAR100, including attacks with different step sizes, $\epsilon$, both $\ell_2$ and $\ell_\infty$ norm. Our result shows that there is indeed a slight decrease in the robustness gain, however, the improvement largely remains and surpasses vanilla models trained with the same defense methods. Hierarchical robustness is a novel concept that captures mistakes over coarse labels. To the best knowledge of the authors, there is currently no related work (or SOTA) on similar topics. Since the visual separability between groups of fine labels can be highly uneven within a given dataset, and thus some coarse labels are more difficult to distinguish than others. This motivated us to first train a classifier that focuses on distinguishing perturbations at the coarse-level, i.e., distinguishing vehicles from animals. Then, within the coarse classes, we robustify classifiers that focus on distinguishing visually similar inputs. Intuitively, this two-step training procedure of HAR benefits from a single robustified CNN with improved robustness between coarse classes, and multiple robustified FNN with improved robustness between visually similar fine classes.

---

### Official Review · AnonReviewer4 · 2020-10-27
**Paper could benefit from better baselines and more thorough evaluation**

**Rating:** 4
**Confidence:** 4

**Review:**

The authors discuss a new notion of adversarial robustness, specifically, robustness to hierarchical adversarial examples. This is motivated by the idea that some types of misclassifications (e.g. mistaking one type of dog for another) may be less harmful than others (e.g. mistaking a dog for a truck); thus, adversarial examples will be more harmful if they cause coarse-grained errors as opposed to fine-grained ones. The authors further propose a network architecture that is designed to be more robust in the hierarchical sense, and they provide some experimental evidence comparing their methodology to standard adversarially trained networks.

I am in favor of rejecting this paper because it does not provide adequate baselines or proper and thorough robustness evaluation, which is especially important for the field of adversarial robustness.

The main strength of this work is the idea to analyze hierarchical adversarial examples. The reasoning for studying this is sound, and using CIFAR-100 to do so seems quite reasonable (The fact that the authors focused on CIFAR for their work is fine for me. If the authors wish to also do experiments on ImageNet, hierarchies based on ImageNet can also be created based on https://wordnet.princeton.edu/, for example, as in https://arxiv.org/abs/2008.04859.)

However, the paper has weaknesses in terms of its baselines and evaluation. The main idea of studying hierarchical adversarial examples is clear, but is simple, so it would be good to have comprehensive evaluation to help us understand how different architectures, different training methods, and different Lp-norms and Lp-epsilon values affect this property. The authors only present results that vary the training method and they try two architectures (a standard ResNet34 and their proposed HAR architecture), but they do not study different Lp norms, epsilons, or a larger number of standard architectures.

Next, it is not surprising that standard models, which are trained to be robust to standard (fine-grained) adversarial examples, are less robust to coarse-grained adversarial examples compared to HAR, which is specifically designed with the task of coarse-grained adversarial examples in mind. Instead, I would have liked to see a simple adversarial training baseline that involves creating coarse-grained adversarial examples during the adversarial training phase. Otherwise, it is hard to understand how much the HAR architecture itself is important (as opposed to focusing on coarse-grained adversarial robustness rather than fine-grained adversarial robustness during training). Finally, HAR is actually worse than standard networks (trained for fine-grained robust classification) on most evaluation metrics except for “targeted” coarse attacks, where HAR shows some improved robustness.

Next, evaluating robustness to adversarial examples in general requires much more rigor than is presented. This is a particularly problematic aspect of the robustness field, as pointed out in https://arxiv.org/abs/1802.00420, https://arxiv.org/abs/1902.06705, and https://arxiv.org/abs/2002.08347. One example of a change is that the authors could perform a much stronger attack than PGD20 (e.g. PGD200 or PGD1000, and ideally with random restarts), in addition to performing the sanity checks discussed in https://arxiv.org/abs/1902.06705, before claiming improved empirical robustness.

Minor comment:

My understanding was that generating worst-case targeted hierarchical adversarial examples takes a long time, because it requires iterating over all possible fine-grained classes. Is there a way to make this faster? A faster way to do this would make it more feasible to do standard adversarial training, where coarse-grained adversarial examples are generated during the training.


Post Rebuttal Update:

I have read the author's rebuttal, and appreciate the additional experiments. I will maintain my score due to the importance of extensive evaluation for showing empirical robustness, and a lack of proper baselines. In particular, I think that some version of "adversarial training with coarse labels" should be better than standard adversarial training in terms of hierarchical robustness; only after that can we evaluate the benefits of HAR in comparison to "adversarial training with coarse labels."

---

> ### Author Response · Authors · 2020-11-25
> **Response to the Reviewer**
>
> Thank you for the careful reading of the paper and your helpful comments. We revised the paper with an extensive empirical study on CIFAR100, including attacks with different step sizes, $\epsilon$, both $\ell_2$ and $\ell_\infty$ norm. Also, all PGD adversaries are randomly initialized. We also included a modified PGD adversarial training (ADV-T) with adversaries generated in a similar, targeted fashion. In particular, given an input pair from the training set $(x,y)$ and $y \in z^*$, the perturbation is computed based on a targeted PGD attack where the target label is uniformly random sampled from {$y \mid y \not \in z^*$}. Note that ADV-T only applies to vanilla models. We observed that vanilla models trained using ADV-T shows improved robustness against untargeted PGD attacks compared to the original adversarial training method. However, we noticed that there is a significant decrease in the hierarchical robustness of ADV-T models against the hierarchical adversarial examples proposed in Section 2.1, leading to a worse hierarchical robust accuracy compared to ADV. Lastly, we emphasize that since evaluating robustness empirically using attacks only shows an *upper bound* on the true robustness of the model, so one should focus on the metric that results in the *lowest* upper bound. For this reason, a network outperforming HAR on weaker metrics does not imply it has better hierarchical robustness. We agree with the reviewer that the current design of the worst-case targeted attack is not suitable for adversarial training (thus the random label selection in ADV-T). One possible candidate is to generate a targeted attack based on an intra-epoch evaluation on a hold-out set.

---

### Official Review · AnonReviewer2 · 2020-10-27
**Review of "Improving Hierarchical Adversarial Robustness of Deep Neural Networks"**

**Rating:** 4
**Confidence:** 3

**Review:**

# Summary
This paper investigates adversarial attacks in the hierarchy of labels. Misclassifying a person for a car is a “coarse misclassification”, but misclassifying a bus for a car is a “fine misclassification”. The paper introduces both a metric for hierarchical adversarial robustness, and a method to improve this metric (which is a hierarchical formulation of resnet architectures).

# Strong & Weak points
## Strong points

  * This paper addresses the problem of hierarchical adversarial examples, which is a phenomenon most often ignored in related literature. Adversarial attacks, like PGD, often yield classes in the same coarse class.
  * Table 2 shows that the Hierarchical formulation of the network improves robustness against worst case targeted adversarial attacks.

## Weak points

  * The paper relies on experimental results with small datasets, like CIFAR10 and a subset of CIFAR100. Moreover, the ImageNet dataset is mentioned already in the second paragraph and the frequent citation “Deng 2012” also uses ImageNet. So I am surprised that this paper does not cover this dataset. If this paper aims to focus future research on hierarchical adversarial robustness, I suggest including at least the definition of coarse labels for the ImageNet and OpenImages datasets.
  * It is unclear to me how the adversarial class is defined. To quote from the paper “ In our work, we consider a worst-case attack by iterating through all eligible target labels until a successful misclassification occurs, and the attack is unsuccessful when the target list is exhausted.” The definition of a worse-case attack is key to the proposed method, and necessitates a clear explanation.

# Statement
Recommendation: 4
Reasons

  * Results on small datasets like CIFAR10 and a subset of CIFAR100, while the ImageNet is mentioned and cited multiple times.
  * Motivation is not backed by data or citations. The paper hinges on the claim that PGD attacks yield misclassifications within the same coarse label, but this claim is not substantiated with data or citations.

# Supporting arguments

  * I suggest the paper gets grounded in more relevant literature, or provide more extensive experiments. The paper contains only two experiments (table 2 and table 3), has no “related work” section, and has nor 23 cited works. I’m aware that the ICLR conference places no restriction on the structure of the paper, or the number of experiments, but I suggest expanding either the experiments or the related work to increase the arguments in the paper.
  * The paper misses numbers to quantify the motivation for this research. The contributions even state that “We find that a vast majority of the misclassifications from the untargeted attack are within the same coarse label”, but this statement is not backed by data in the rest of the paper. The proposed method is only useful when PGD attacks indeed yield misclassifications within the same coarse label, so quantifying this phenomenon will provide a strong argument for adoption of the new metric.

# Questions

  * The paper defines a new subset of CIFAR100, named CIFAR100-5x5. What were the design choices for this new dataset and why?
  * In what light to ResNet10 and ResNet34 have the same capacity? To quote from the paper “We use models with a lower capacity so that both vanilla models and the hierarchical classifiers have the same order of magnitude of parameters”. Could you please elaborate on the calculation?

# Minor feedback
These points are not part of the assessment

  * When defining $F(x) : R^d \arrow R^n$, the variables $d$ and $n$ are not defined. Moreover, when defining the dataset, $\{ x, y \}_n$, the variable $n$ is reused in a different context.
  * Please be consistent in the mathematical expressions. On page 2, the prediction is defined as $\arg \max_i F(x)_i$, on page 3, the prediction is defined as $\arg \max F(x)$, and later on page 3 the prediction is indicated with lower case, $f(x)$.
  * Table 2 (and its reference in the main text) consider “performance” of the model. Please either a) use terms such as accuracy/error-rate to indicate what the numbers represent, or b) write in the caption whether a higher score or lower score indicates better performance.

---

> ### Author Response · Authors · 2020-11-25
> **Response to the Reviewer**
>
> Thank you for the careful reading of the paper and your helpful comments. To generate a worst-case attack for a training input pair $(x,y)$ where $y\in z^*$,  we perform a targeted PGD attack and iterate through all fine labels in {$y \mid y \not \in z^*$}, i.e., fine labels which do not belong in the same coarse class. This iterative attack process terminates if 1. perturbation results in a successful targeted misclassification; 2. all candidate fine labels have been used as targets. We will include an algorithm box to help clarify the definition. The observation that a vast majority of the misclassifications from the untargeted attack are within the same coarse label can be seen in Table 1. Notice that under untargeted PGD attacks, most models have lower "fine" accuracy compared to "coarse" accuracy. This suggests that more mistakes are being made within the coarse label. We revised the paper with an extensive empirical study on CIFAR100, including attacks with different step sizes, $\epsilon$, both $\ell_\infty$ and $\ell_2$ norm. ResNet10 and ResNet34 do not have the same capacity, however, multiple ResNet10's in HAR leads to a similar capacity compared to ResNet34.

---

### Official Review · AnonReviewer1 · 2020-10-28
**Important problem, but lacking baselines and detailed experimental evaluation**

**Rating:** 5
**Confidence:** 4

**Review:**

Summary: This paper tackles the problem of building hierarchical adversarially robust (HAR) models---i.e., models that are less prone to coarse-grained misclassifications in the face of adversarial manipulation. Specifically, the authors propose HAR networks, wherein the learning/inference problem is decomposed into coarse-grained classification followed by a number of fine-grained classifications. The proposed approach is evaluated on CIFAR-10 and (a subset of) CIFAR-100 for linf adversarial attacks.

Comments: The problem studied in this paper is a natural and pertinent one. The paper is also fairly well-written. The proposed approach does seem to offer some gains in coarse targeted accuracy, albeit occasionally at the cost of fine accuracy.  However:

* One thing that is not entirely clear from the paper is how exactly the authors perform the worst-case targeted attack. The description in Section 2.1 is vague and the authors should clarify. From what I understand, the authors maximize the loss with respect to every fine-grained label belonging to a different coarse class and pick the worst. How exactly do the authors determine the loss with respect to fine-grained labels from other coarse classes? What happens if you instead do an *untargeted* attack on the *coarse classifier* alone? This exactly corresponds to the quantity of interest and the authors should include an evaluation based on this attack.

* The authors do not justify why they choose separate networks for the coarse and fine classification tasks. One could imagine training a single flat model with a hierarchical robust loss. This idea has been explored in prior work for standard models---for instance, YOLO 9000 does this for ImageNet https://arxiv.org/abs/1612.08242. Does this perform worse than the HAR architecture proposed in the paper? Overall, training a flat classifier with a hierarchical loss seems like a more efficient and scalable approach for larger datasets that may have many levels of coarse classes.

* I also think the claims in the paper would be stronger with additional evaluation. Specifically: (i) The authors should show that these trends hold for a few different eps and for l2-robust models as well. It would also be valuable to validate these results on all of CIFAR100 or even subsets of ImageNet (which may have more than one level of coarse labels). (ii) As a sanity check, the authors should evaluate the robustness of their models (trained with linf eps=8/255) as a function of inference time eps ([0, 1]) and PGD steps. Further, the authors should evaluate black box accuracy of their models.

Minor:
- Section 2.1 typo: “none-true target label”.
- Section 4.1 typo: “traininig”.

Overall, I think this paper studies an interesting problem, but could be improved in terms of experimental evaluation and comparison to baselines (as described above). I also have concerns about the scalability of this approach to larger datasets with a multi-level hierarchical structure. I would be happy to increase my score if the authors address my concerns.


### Post-Rebuttal Update ###

I thank the authors for their response and edits to the paper. In particular, the authors have tried to address comments from the reviewers pertaining to further empirical evaluation of their approach. However, I still have two concerns about the paper post-rebuttal:

[Baselines] I share Reviewer 4's concern that the authors do not compare to several natural baselines. As I mentioned in my review, I am still not convinced about the design choice of using multiple networks for each set of coarse/fine-grained labels. There are a host of approaches (both training losses and architectures) to deal with hierarchical classification in the standard setting. The authors do not justify quantitatively why adapting these methods to the robust setting (i.e., incorporating ADV/TRADES there) will not work. Further, I do not find the justification of fine-grained labels within a dataset being uneven for using multiple networks to be sufficient . This seems like a more fundamental problem that should be fixed by changing the dataset/class hierarchy. In fact, having even coarse/fine classes is essential to justify the merits of HAR in the first place.

[Evaluation] Based on the new results added to the paper in the rebuttal phase, it seems that HAR does worse than vanilla ADV on coarse accuracy under untargeted attacks (Table 1); while vanilla ADV  does worse than HAR using worst-case attacks (Table 2). If we compare the minimum over the two attacks (as is standard to correctly measure robustness), the coarse robustness is actually *better for vanilla ADV than HAR* (24.60% vs 20.71%). This finding seems to go against the main claim of the paper.

[Other] There seems to be a discrepancy in Table 1---the robust accuracies go up between the PGD20 eval and PGD50 eval.

Based on the concerns listed above, I will maintain my original score.

---

> ### Author Response · Authors · 2020-11-25
> **Response to the Reviewer**
>
> Thank you for the careful reading of the paper and your helpful comments. The *worst-case* nature of the proposed hierarchical attack in Section 2.1 is that we perform a targeted PGD attack and iterate through ALL possible fine labels, i.e., fine labels which do not belong in the same coarse class. This iterative attack process terminates if 1. perturbation results in a successful targeted misclassification; 2. all candidate fine labels have been used as targets. Thus, we do not compare losses. We use the *worst-case* descriptor to differentiate the proposed attack from *average-case* and *best-case* attacks, which we discuss in detail in Section 2.1 We included an algorithm box to better illustrate the proposed method. For evaluation of untargeted attacks on the coarse classifier alone, we discuss this in Section 4.3, where we generate untargeted attacks based on the coarse classifier of HAR. Our result shows that such attacks are not as strong as the hierarchical attacks we introduced in Section 2.1.
>
> The method described in the YOLO is for a rather different goal. They leveraged the hierarchical structure of a larger dataset to combine two separate, but related datasets. Previous work has shown that the YOLO and other object detection algorithms are vulnerable to attacks (Zhao, Yue, et al, 2019). Since the visual separability between groups of fine labels can be highly uneven within a given dataset, and thus some coarse labels are more difficult to distinguish than others. This motivated us to use separate networks for the coarse and fine classification tasks.  Intuitively, HAR benefits from a single robustified CNN with improved robustness between coarse classes, and multiple robustified FNN with improved robustness between visually similar fine classes.
>
> Lastly, we revised the paper with an extensive empirical study on CIFAR100 including attacks with different step sizes, $\epsilon$, both $\ell_\infty$ and $\ell_2$ norms.
>
>
> Reference:
> Zhao, Yue, et al. "Seeing isn't Believing: Towards More Robust Adversarial Attack Against Real World Object Detectors." Proceedings of the 2019 ACM SIGSAC Conference on Computer and Communications Security. 2019.

---

### Decision · Program_Chairs · 2021-01-07
**Final Decision**

**Decision:**

Reject

**Comment:**

This work targets an important problem: susceptibility of ML models to adversarial perturbations that make them completely misclassify an input, as opposed to "just" fail to get the right fine-grained class while getting the correct coarse-grained one. This natural question did not receive enough attention so far, so having this work look into it is a definite plus.

However, as the reviewers point out, this study has a number of issues in terms of the methodology of the experiments. For example, it is unclear whether the proposed (natural) variant of training the robust model is particularly beneficial for the stated goal. As such, it seems that the paper is not ready for publication and the authors are strongly advised to revise the article and submit it again.